



# Ground-based observations of periodic temperature fluctuations in the mesopause region with periods larger than 2 days

Christoph Kalicinsky[1,*], Robert Reisch[1,2,*], and Peter Knieling[1]

[1]Institute for Atmospheric and Environmental Research, University of Wuppertal, Germany
[2]Institute of Climate and Energy Systems: Stratophere (ICE-4), Research Center Jülich, Germany
[*]These authors contributed equally to this work.

**Correspondence:** Christoph Kalicinsky (kalicins@uni-wuppertal.de)

**Abstract.** We analysed more than 30 years (1988–2021) of OH(3,1) rotational temperatures observed from Wuppertal, Germany, with respect to periodic fluctuations (2 to 60 d) using the Lomb-Scargle periodogram. The main type of fluctuation observed in the last decades shows a period of about 28 d and is most likely a Rossby wave (1,4) mode. Other periods which are frequently found in the observations lie in the period ranges 5 to 6 d, 8 to 12 d, and around 15 d and can likely be assigned

to the quasi-5-day, the quasi-10-day , and the quasi-16-day wave, respectively. According to theory, these observations are the Rossby wave (1,1) mode, the (1,2) mode, and the (1,3) mode, respectively.

The wave activity is typically larger in winter time than in summer time because of the different wave filtering in summer and winter. This winter to summer difference holds for waves with larger periods, but it breaks off in the case of smaller periods below about 20 d. The occurrence frequency of these waves exhibit two smaller maxima around the equinoxes.

The long-term behaviour of the wave activity shows a quasi-bidecadal oscillation. A further analysis suggests that the yearly mean amplitude of the significant events shows this oscillation not the number of days with significant events in one year. This means, that in certain years not more events but events with larger amplitudes are expected, whereas in other years the mean amplitude of the events is smaller.

## 1 Introduction

Planetary waves are large scale global phenomena that are known to have an important role for the global circulation due to transport and deposition of momentum for a long time (e.g. Salby , 1984; Andrews et al, 1987; Volland , 1988). They are typically generated at lower altitudes, propagate upwards, and are even able to reach the mesosphere and lower thermosphere (MLT) region under certain conditions (e.g. Holton , 1984; Laštovička , 1997; Smith , 2003; Sassi et al., 2012). Ground-based observations of wind, temperatures, and airglow in the MLT region proofed as a good way to observe planetary or planetary-

like waves with different periods and monitor their temporal evolution (e.g. Wu et al., 1994; Espy et al., 1997; Yoshida et al., 1999; Bittner et al., 2000; Luo et al., 2000; Takahashi et al., 2002; Kishore et al., 2004; Espy et al., 2005; Höppner and Bittner , 2007; Stockwell et al., 2007; Day and Mitchell, 2010a, b; Takahashi et al., 2013; Egito et al., 2018; Zhao et al., 2019). In the MLT region waves with largely different periods have been observed in the past. These periods range from only a few days in the case of very fast waves (e.g. Yoshida et al., 1999; Egito et al., 2018) to periods in the range of almost an week to some





weeks (5 to 30 days) (e.g. Wu et al., 1994; Espy et al., 1997; Luo et al., 2000; Jarvis , 2006; Day and Mitchell, 2010a, b; Takahashi et al., 2013; Zhao et al., 2019) to periods even longer than 30 days (Espy et al., 2005; Stockwell et al., 2007). Some of these fluctuations at specific periods can be assigned to different Rossby wave modes. These modes are influenced by the distribution of zonal background winds and appear in the presence of such winds in rather specific period ranges such as the Rossby wave (1,4) mode at about 28 d and the (1,3) mode at about 16 d (e.g. Kasahara , 1980; Salby , 1981a, b).

Compared to satellite observations which only observe a local point a few times a day, ground-based observations have the large opportunity of measuring nearly continuously. Furthermore, some ground-based instruments have been operating since several decades as the instrument can be maintained all the time which is not possible for satellite instruments. Thus, ground-based instruments and their corresponding long-time records are very suitable not only for the detection of waves but also for the analysis of the long-term evolution and the occurrence frequencies of waves with different periods. The observations by

the GRIPS (GRound-based Infrared P-branch Spectrometer) instruments at Wuppertal exhibit one of the longest temperature time series for the mesopause region around the whole globe which has been used in several different studies (e.g. Bittner et al., 2000; Oberheide et al., 2006; Höppner and Bittner , 2007; Offermann et al., 2010; Kalicinsky et al., 2016, 2018, 2024). The observations at Wuppertal have already been used to analyse planetary waves in the past by using wavelet transform and wave proxies (Bittner et al., 2000; Höppner and Bittner , 2007). Höppner and Bittner  (2007) observed a long-term periodic

behaviour of the wave activity with a period of roughly 20 years similar to the Hale cycle. In their study they used the standard deviation of the temperature residuals (after subtracting the seasonal variations) as proxy for the planetary wave activity as at least a large part of these temperature fluctuations are thought to be caused by planetary or planetary-like wave activity (Bittner et al., 2000; Höppner and Bittner , 2007).

In our new study we now have the large advantage of a much longer time series as Höppner and Bittner  (2007) only used

observations until 2005. Furthermore, we used a different technique which is based on the Lomb-Scargle periodogram to detect the periodic fluctuations and which is well suited to handle time series with data gaps (Kalicinsky et al., 2020). Compared to the previous studies this technique no longer requires data assimilation before the analysis to get rid of the data gaps (Bittner et al., 2000; Höppner and Bittner , 2007). The paper is structured as follows. Sect. 2 describes the measurements technique and the data as well as the method to detect the periodic fluctuations with some improvements made during this study. The

results of the analysis with respect to the occurrence frequencies of waves with different periods and the long-term behaviour of planetary scale waves are presented in Sect. 3. These results are discussed and compared to previous results and different other observations in Sect. 4. Finally, Sect. 5 summarizes the results.

## 2  Measurements and data analysis

This section summarises all important information regarding the measurements and data analysis. First, the instruments and

the measurement technique are described. Second, the detrending of the OH* rotational temperatures by using a seasonal fit is explained. In the last subsection the moving Lomb Scargle method is explained. This method is used to analyse the residual temperatures with respect to periodic fluctuations in the period range between 2 and 60 d. In contrast to Kalicinsky et al. (2020),





where the method is described for the first time, we improved the method in this study to overcome some of the drawbacks of the former method. These improvements are intensively described in Sect. 2.3.

## 2.1 GRIPS instruments

The temperature observations used in this study were derived from the measurements of two GRIPS (GRound-based Infrared P-branch Spectrometer) instruments, namely GRIPS-II and GRIPS-N. GRIPS-II started its measurements in the early 1980s and continuously measured since mid-1987. It stopped working in 2011 because of a detector failure. The GRIPS-N instrument is the follow-up instrument of GRIPS-II and continues the measurements until now (Kalicinsky et al., 2018, 2024). Both instruments were operated at Wuppertal, Germany ($51°$ N, $7°$ E). GRIPS-II is a Czerny-Turner spectrometer with a Ge detector cooled by liquid nitrogen (see e.g. Bittner et al., 2002, for instrument details). The GRIPS-N instrument is also a Czerny-Turner spectrometer equipped with a thermoelectrically cooled InGaAs detector. The instrument has very similar optical properties as the GRIPS-II instrument, which makes it a suitable replacement instrument (Kalicinsky et al., 2018, 2024).

Both instruments measure three emission lines of the OH*(3,1) band in the near infrared region (1.524 µm–1.543 µm), namely the $P_1(2)$, $P_1(3)$, and $P_1(4)$ line. The layer of excited OH molecules is located at about 87 km height and has a layer thickness of approximately 9 km (Oberheide et al., 2006; Offermann et al., 2010). The measurements were carried out every night, except in nights with cloudy conditions. Favourable measurement conditions are given in about 220 nights per year (Oberheide et al., 2006; Offermann et al., 2010). The relative intensities of the $P_1(2)$, $P_1(3)$, and $P_1(4)$ lines are used to derive rotational temperatures (see Bittner et al., 2000, and references therein). The OH* rotational temperature may deviate from the kinetic temperature, especially in cases of higher vibrational states. However, rotational temperatures that were determined from emissions originating from the OH*(3,1) band are expected to be close to the kinetic temperatures (Noll et al., 2015). In addition, the analysis of periodic fluctuations does not require absolute values.

## 2.2 Seasonal variations of OH* rotational temperatures

In order to analyse periodic fluctuations with periods of a few days to a few weeks one has to remove the seasonal variations first as these variations have large amplitudes and otherwise would mask the periodic fluctuations of interest. The seasonal variations can be described by an annual, a semi-annual and a ter-annual cycle (Bittner et al., 2000; Kalicinsky et al., 2024). The full description of the seasonal fit is

$$T = T_0 + \sum_{i=1}^{3} A_i \cdot \sin(\frac{2 \cdot \pi \cdot i}{365.25}(t + \phi_i)), \tag{1}$$

where $T_0$ is the annual average temperature, $t$ is the time in days of year, and $A_i$, $\phi_i$ are the amplitudes and phases of the sinusoids. A fit according to Eq. 1 is used to derive residual temperatures for each year separately. Fig. 1 shows an example for the seasonal variations. The black dots show the observed OH* rotational temperatures and the red curve shows the fit curve. In the lower panel of Fig. 1 the residual temperatures (data - fit curve) are shown. These residual temperatures are used to derive periodic fluctuations.



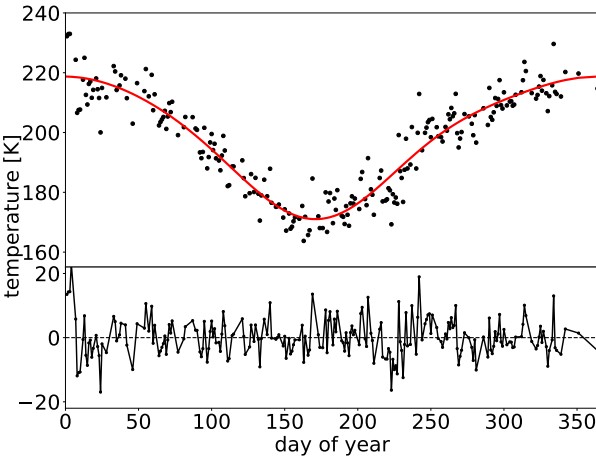

**Figure 1.** Seasonal variations of the OH* rotational temperatures in the year 1997. The black dots show the OH* rotational temperatures and the red curve shows the fit curve according to Eq. 1. In the lower panel the residual temperatures (data - fit curve) are displayed.

### 2.3    Moving Lomb-Scargle periodogram

The periodic fluctuations are analysed with the moving Lomb-Scargle periodogram (Kalicinsky et al., 2020). This method is based on the original Lomb-Scargle periodogram (LSP), a method that can detect periodic fluctuations in all kind of time series even in time series with unequal spacing, which is the case when data gaps are present (Lomb , 1976; Scargle , 1982). Thus, it is very suitable to analyse OH*(3,1) rotational temperature time series that exhibit irregular data gaps due to weather conditions. In the approach described by Kalicinsky et al. (2020) a time window of fixed length, for example 60 d, is used for the analysis.

The starting point is the beginning of the time series and the window is shifted by the minimum sampling step (here one day) until the end of the time series is reached. For the data points within each of these individual time windows, that all include a different part of the complete time series, a LSP is calculated separately. In this way periodic fluctuations can be detected as well as the time evolution of the periods and amplitudes of these fluctuations can be observed (see Kalicinsky et al., 2020, for a complete description of the method).

By contrast to Kalicinsky et al. (2020), in this study the analysis is performed such that the centre days of the time windows cover a complete year of observations. Because of the window size, which is, for example, 60 d, the seasonal fit (compare Eq. 1) has to be calculated for a time interval larger than one year. Only then also results for time windows centred around the beginning and the end of the year can be determined. Notice here, that these results then include observations from the previous and following year, respectively, but each possible centre day is only used once and all possible time windows have

been of the complete time series were analysed. In this study we extended the time interval that is used for the seasonal fit by two months at both sides. Another benefit of this approach is a better constrained fit at the beginning and the end of the year, since the behaviour of the observations before and after the year of interest is considered. Therefore, the new fit for the larger time interval typically compares better to a fit that is calculated for the complete winter only (fit time interval: 01.07. – 30.06.)

 

at the beginning and end of the year.

The use of a fixed window length has also drawback, which is mainly noticeable in the case of signals with periods much
smaller than the window length. It is not very likely that such a signal with a period of only a few days remains in atmospheric
temperatures for several weeks. When the time window is much longer than the duration of such an event the results using
the LSP are damped, i.e. the period and amplitude is a mean value over the whole time interval and typically much smaller
than it would be when only the shorter duration length of the event would be used as time window. In the worst case events

with small periods and a small duration may be missed during the analysis. In order to avoid this drawback we introduced
a varying window length here. The window length changes in the same way as the analysed period, i.e. a smaller window is
used for smaller periods and vice versa. A minimum value for the window length and thus the period can be chosen below
which the window length stays constant. But this minimum length of the used window has also restrictions. As the significance
of the results depends on the number of data points (e.g. Cumming et al. , 1999; Zechmeister and Kürster , 2009; Kalicinsky

et al., 2020), it is not possible to use too small windows. In the case of more data points the significance level is lower, i.e.
the variance explained by the analysed sinusoid can be lower than it would have to be the case when less data points were
analysed. Therefore, the same explained variance could be significant when more data points were incorporated in the analysis
and insignificant in the case of less data points. In total, the minimum window length has to be a trade-off. On the one hand, it
has to be large enough to avoid problems with too many data gaps and worse significance compared to a larger window and,

on the other hand, it has to be small enough to detect events with smaller periods that occur in the time range of days to one
or two weeks. We also introduced a factor that defines the relation between the window length and the period that is analysed
with this window length. This value can be seen as a factor that defines the number of cycles of an oscillation that fits in the
time window at a given period. A factor of one means the analysed period and the time window are always the same. In the
case of the factor of 2 the window length is twice as large as the period analysed using this window. The minimum window

length (divided by the factor) still defines the minimum period below which the time window stays constant. For example the
scaling factor of 2 sets the minimum period to 12 d up to which the periods below are analysed with the defined window length
of 25 d. After the period of 12 d the rounded double window length corresponding to the analysed period is used, e.g. 26 d for
12.6 d, 27 d for 13.2 d, 30 d for 14.9 d, 40 d for 19.9 d and 120 d for 60 d. The scaling of the minimum period has been done
to ensure a softer transition to larger window lengths.

Compared to the former approach described in Kalicinsky et al. (2020) the range of used window lengths is now larger, since
the window length changes in relation to the analysed period. In the former evaluation of the significance in dependence of
the window length only lengths of 30 d and larger have been considered. In this study we also use smaller window lengths.
Therefore we reevaluated the significance in dependence of the used window length and the analysed period range in a larger
range of window lengths (10 to 90 d) and for more representations. Here we simulated 100 times 10000 representations instead

of only one time 10000 representations as done in Kalicinsky et al. (2020). In this way we obtained new results that give
refined coefficients for Eq. 6 in Kalicinsky et al. (2020) with an improved estimation of the uncertainties. The new values of
coefficients are $2.98 \pm 0.02$ dd$^{-1}$ for the slope and $-0.23 \pm 0.01$ d$^{-1}$ for the intercept. The given uncertainties are the one
times the standard deviation of them mean. Thus, the new results show that the former results by Kalicinsky et al. (2020) were





by chance at the end of the $3\sigma$ range for both values. Additionally, small differences may occur from the wider range used for
the window lengths.

Fig. 2 shows a comparison between the new results using the varying window and the old results with a fixed window length of
60 d for the example year 2005. The values used for the analysis with varying window length are a minimum window length
of 25 d and a factor of 1.0. The window length stays constant at a period of 25 d and below and monotonically increases with
increasing period above this value. Obviously, the new results allow for more variability of the amplitude as the smoothing
effect is smaller. Because of these less fluctuations and also due to better significance levels in the case of more data points
used for the analysis, it also may occur that the time interval showing significant results is slightly larger in the time domain
for the results with the fixed window. This effect mainly shows up at larger periods. The large benefit of the new method arises
at smaller periods. Here in Fig. 2a, thus the result for the improved method, many more significant events are detected. These
events are typically very short in duration. In the case of a fixed window with too large size these events will be smoothed
such as they are not detectable any more (see Fig. 2b). Thus, with the method described by Kalicinsky et al. (2020) significant
events could be missed and the improved method presented here gives a large benefit.

## 3 Results

This section is divided into two parts. In subsection 3.1 the periods of different fluctuations that are typically caused by
planetary waves are analysed. This includes the occurrence frequencies of the observed periods and also seasonal differences
of the occurrence frequencies for specific period ranges. Subsection 3.2 focuses on the long-term evolution of the wave activity,
which mainly deals with the question if there is a long-term trend or even a long-term periodic behaviour of the wave activity.
All presented figures were deduced from LSP results that have been calculated with the new approach with varying window
length. The minimal window length was 25 d and the factor 1. Only periodic fluctuations that have been classified as significant
events were taken into account, i.e. only results inside the contour lines in Fig. 2a were used.

### 3.1 Occurrence frequencies of waves

There may be a drawback when only the event as a whole is considered and counted. Possibly, this would bias the results to-
wards events with a short duration that occur very often. On the other hand, long lasting events may be incorrectly represented
in the final results. In order to avoid this imbalance, we counted the days of each individual event, i.e. all centre days which lie
inside a contour line and, thus, show a significant event. These days with significant results are assigned to the corresponding
period which shows the maximum amplitude. By counting in this way the occurrence frequency becomes larger in two situa-
tions. First, it is larger when short events occur very often and, second, the occurrence frequency increases when events have a
long duration. Both situations are, in our opinion, equal in importance. As events with larger period tend to also last longer, it
is expected that there is an overall increase of the counted days with increasing period.

Figure 3 shows the occurrence frequency of events with certain periods. In total, one can see the expected overall increase of
the occurrence frequency with increasing period (compare Fig. 2a). Nevertheless, at some specific periods events occur more




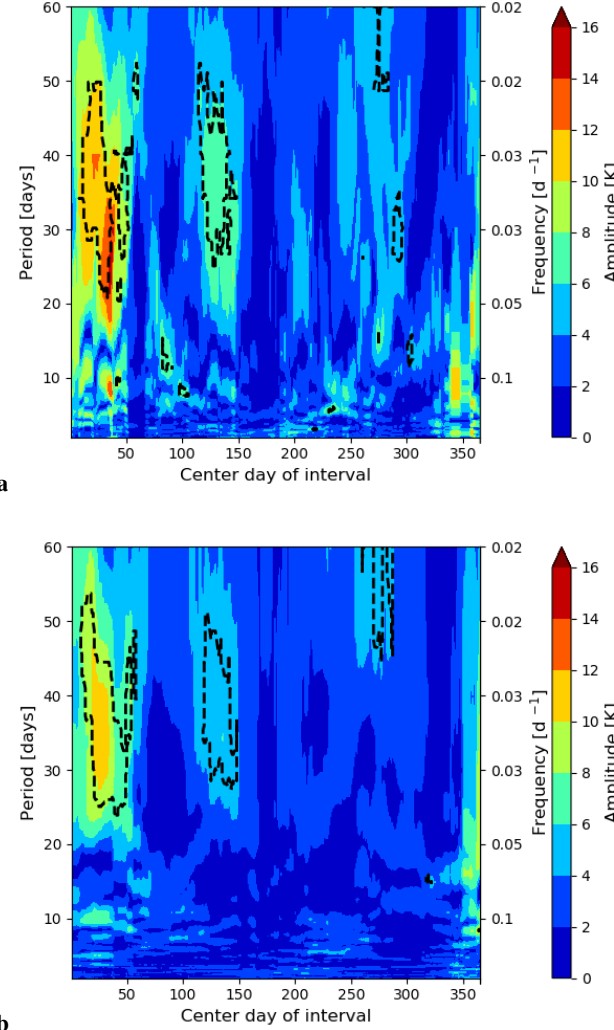

**Figure 2.** Comparison of the LSP results for the amplitude in the year 2005. **a** Improved method with varying time window. The minimum window length used for the analysis was 25 d, i.e. below a period of 25 d the window length was constant at 25 d an monotonically increased above this threshold period. **b** Former method with fixed window length of 60 d. The x-axes show the centre days of the time windows and the y-axis the period and frequency, respectively. The amplitude displayed is colour coded. The black contour lines mark the region of significant results.

often than events at other periods. Here, the counted days clearly separate from the expected overall increase of these days with increasing period. These preferred periods are about 28 d, 15 d, 12 d, 10 d, 8 d and in the period range of 5 to 6 d.

The seasonal dependency of the occurrence frequencies is largely dependent on the period itself. Figure 4 shows the number of centre days with significant events plotted against the month for different period ranges. In the upper row of Fig. 4 the period



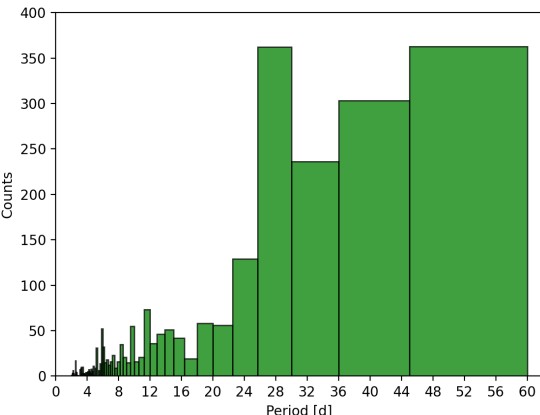

**Figure 3.** Occurrence frequency of events in dependency of the period. The histogram shows the number of centre days of the LSP analysis with significant results plotted against the corresponding period.

ranges are larger than 10 d, larger that 20 d, and larger than 30 d (left to right). Obviously, events with larger periods more often occur in winter time than in summer time, whereby the largest numbers were observed in January and February and the lowest numbers in June. In the case of events with small periods the situation is largely different. The lower row in Fig. 4 shows the same type of plots for the period ranges below 10 d, below 20 d, and below 30 d (left to right). In the period ranges including only the events with the smallest periods the largest part of the events were detected around the equinoxes (late March and

late September). The maximum number of significant events occur in April and September. Thereby, the number of events observed around the equinox in fall is larger than in spring time. Note here, that this important result is only visible by using the new approach with the varying window length. When using the former approach described in Kalicinsky et al. (2020) a large number of events with smaller periods are missed and the clear seasonal structure with the maxima around the equinoxes is obscured. When all events with a period below 30 d are considered, these maxima around the equinoxes diminish and the

large maximum in winter (January and February) dominates. The explanation is that in this period range the large number of events with a period of about 28 d are considered (compare Fig. 3) and these events mainly take place in winter.

The distribution of the observed amplitudes of the events is shown in Fig. 5. Note here that all single amplitudes that correspond to centre days within an event are displayed, i.e. one event would have its own distribution of amplitudes. The mean amplitude of all events is 7.9 K with a standard deviation of 2.8 K. The smallest amplitudes that belong to a significant

fluctuation are slightly above 2 K and the largest observed amplitudes are nearly 16 K. When the amplitudes are plotted for different period ranges (not shown), no obvious difference can be seen. Thus, the amplitudes are not significantly larger for events with smaller or larger periods.



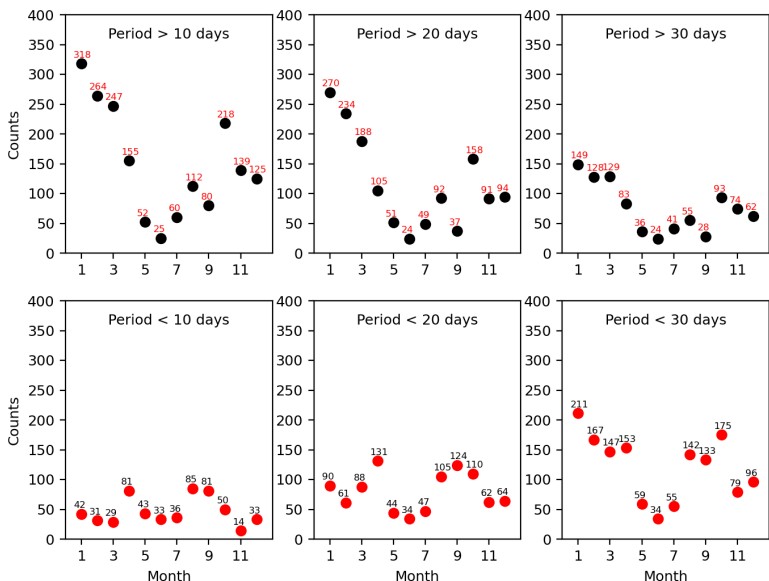

**Figure 4.** Seasonal dependency of the occurrence frequencies of events with different periods. In the upper panel of the figure the number of centre days with significant results are plotted against the month for the period ranges $> 10$ d, $> 20$ d, and $> 30$ d (left to right). In the lower panel the period ranges are $< 10$ d, $< 20$ d, and $< 30$ d (left to right).

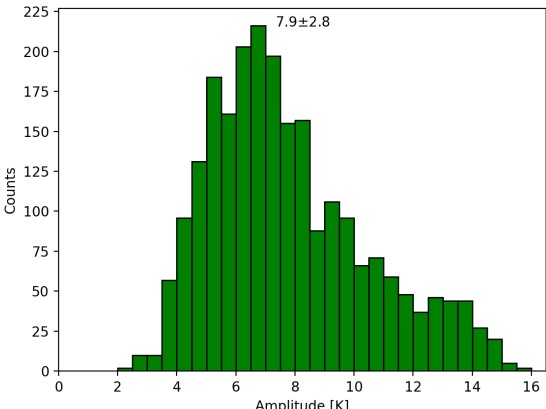

**Figure 5.** Distribution of the observed amplitudes. All single amplitudes that correspond to centre days within an event are displayed.





## 3.2 Long-term evolution of wave activity

The second part of this study deals with the long-term evolution of the wave activity. In former studies this long-term evo-
lution was analysed by using the standard deviation of the residual temperatures (data - seasonal fit) for each single year as
a proxy for the wave activity in this year (e.g. Höppner and Bittner , 2007). The standard deviation is larger when more and
larger fluctuations occur. One possible drawback of this proxy is that the standard deviation includes all kind of fluctuations
independent of the period and significance. Thus, we introduce two new quantities here that are based on the results of the
moving-LSP method. Thus, these new quantities only include significant events. The first quantity is the mean amplitude of the
significant events in one year and, therefore, a measure of the strength of the events. As the wave activity can also be seen as
larger when events occurred more often in a year, although their amplitudes were smaller, we introduced a second quantity that
accounts for this. This quantity is the product of the mean amplitude times the number of days at which a significant event was
detected. Thus, the weighted sum of significant days is used as last proxy. Figure 6a shows the time series of all three different
quantities (the mean was subtracted before). The periodicity of the time series is analysed with the LSP and the results are
shown in Fig. 6b with the same colours as used in Fig. 6a. The two time series in the two upper panels in Fig. 6a, the standard
deviation and the mean amplitude, show a rather similar behaviour with corresponding times of maxima and minima. Only at
some points, for example in the years 2007 and 2009, there are larger differences. The LSPs for these two time series show
also peaks at very similar periods (see Fig. 6b). The main peak in the long-period range is located at about 20 y in both cases,
whereby the maximum in the LSP for the mean amplitude lies slightly below 20 y (red curve in Fig. 6b) and in the case of the
standard deviation slightly above this value (black curve in Fig. 6b). Due to the uncertainty (the width of the peaks) this differ-
ence is not significant. The time series of the weighted sum of significant days does not show the same long periodic fluctuation
as the other two quantities (lower panel of Fig. 6a and blue curve in Fig. 6b). The difference between the time series of the
mean amplitude and that of the weighted sum is simply the number of days with significant results. Consequently, the fact that
only one time series shows the long-term oscillation implies that only the amplitude of the events shows these long-periodic
behaviour and not the number of days. As the standard deviation is a measure of the amplitudes of all fluctuations within the
analysed time interval, the standard deviation shows almost the same long-term behaviour. Furthermore, this also indicates that
in most years the significant fluctuations dominate the standard deviation of the residual temperatures.

## 4 Discussion

The discussion is divided into two parts. First, the possible origin of the observed fluctuations is discussed and compared to
other results. Additionally, the seasonality of the wave activity is examined. In the second part, the long-term behaviour of the
wave activity and the fluctuation of this activity itself is discussed.





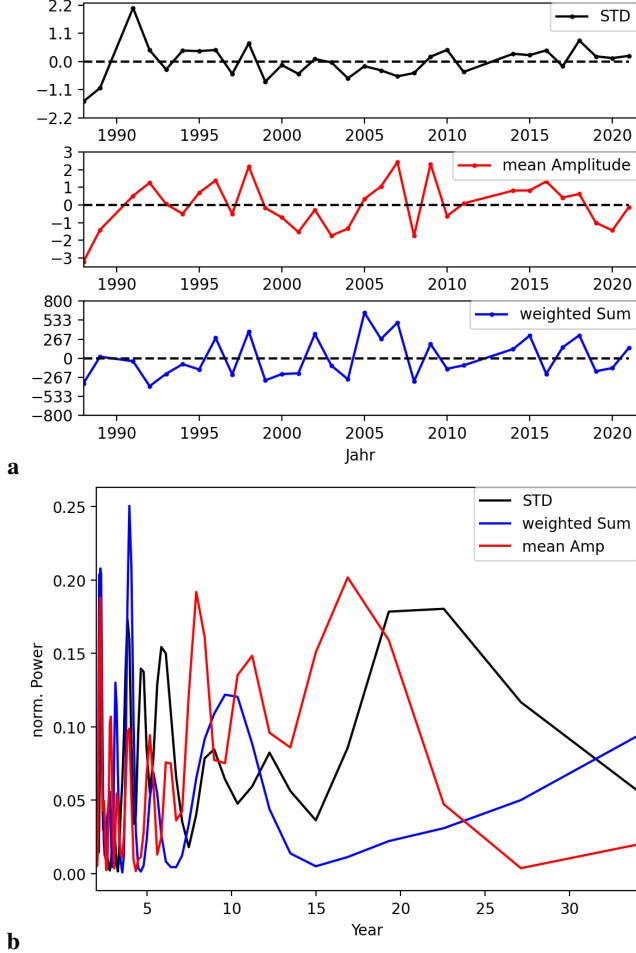

**Figure 6.** Long-term behaviour of the wave activity. **a** The upper panel shows the time series of the yearly standard deviation of the temperature residuals after subtracting the seasonal fit. In the middle panel the time series of the mean amplitude of all significant events in a single year is displayed. The lower panel shows the product of the mean amplitude and the sum of the centre days with a significant result in the LSP. **b** The LSP for the three time series are shown with the same colours as of the time series themselves (black: standard deviation, blue: weighted sum, red: mean amplitude).

## 4.1 Periodic fluctuations

The highest peak in the histogram occurs at a period of around 28 d, which shows that period fluctuations in this period range are very common. And indeed, in a larger number of winters an event with an average period in this range takes place. Fig. 7
shows an example for such an event. The time series of the temperature residuals over the winter 1997/1998 is shown as black curve in Fig. 7a. Obviously, between the days of year (DOY) of about 160 and 270 a large periodic fluctuation is present. The amplitude of this fluctuation increases at the beginning and reaches maximum values around DOY 215. Then it decreases again.



The red curve shows a sinusoid with varying amplitude fitted to the data. A very similar event was observed over Antarctica at Rothera station in the winter 2014 (Zhao et al., 2019). The authors used ground-based and satellite data to analyse the vertical

and horizontal structure of this event and find that it was consistent with the Rossby wave (1,4) mode. This is also consistent with theory where the Rossby wave (1,4) mode has a period of about 28 d in the presence of zonal background winds (Kasahara , 1980). Zhao et al. (2019) state in their study that the propagation in the MLT region is severely limited in summer because of the low phase speed of this type of waves. They also refer to a study by Sassi et al. (2012) which found that the propagation to higher altitudes is expected to be transient because of the effect of the background winds and the wind filtering. This likely

explains the fact that we observe the 28 d wave typically in winter and not in summer. There is another possible effect on the temperature in the mesopause region that shows a periodic behaviour in the right period range: the 27-day solar cycle. But the known amplitudes of this influence are way smaller than the ones observed here. Beig et al. (2008) reviewed studies concerning the influence of the 27-day solar cycle on temperatures in the mesosphere and lower thermosphere and reported values less than 4 K and typically smaller values in the mesopause region than in the mesosphere itself. Different other studies also show

periodic fluctuations in coincidence with the 27-day solar cycle (a time-lag is present) with amplitudes smaller than 1 K (von Savigny et al., 2012; Thomas et al., 2015; Rong et al., 2020). Thus, the amplitudes of the fluctuations observed here are much larger than what is expected in the case of the 27-day solar cycle. We also analysed the GRIPS OH(3,1) rotational temperatures with respect to the influence of the 27-day solar cycle and observed similar amplitudes ($< 1$ K) than in the studies before (von Savigny et al., 2025). Furthermore, we excluded time intervals in winter with large wave activity in the 27 d period range from

the analysis and observed no significant differences to the results before. This led us to the conclusion that the influence of the 27-day solar cycle on the temperatures and the fluctuation with periods of about 28 d that partly show amplitudes larger than 10 K are two completely independent phenomena. Therefore, we believe that the Rossby wave (1,4) modes is the likeliest explanation for the observations in the period range around 28 d.

Another period range that is frequently observed for planetary waves in the MLT region lies in the region around 16 d (e.g. Espy et al., 1997; Luo et al., 2000; Jarvis , 2006; Day and Mitchell, 2010b; Takahashi et al., 2013). Waves with this periods are likely the observation of a Rossby wave (1,3) mode (Kasahara , 1980; Salby , 1981b; Espy et al., 1997). Obviously, the peak in the histogram of our observations (see Fig. 3) is located at about 15 d and, thus, at a slightly smaller period. However, it is still in the range of periods that were predicted in the presence of zonal background winds which range from 12 to 20 d (Salby ,

1981b) and 16 to 19 d (Kasahara , 1980), respectively. Thus, although our observations peak at about 15 d, the observed period range between about 14 to 20 d is still likely connected to this Rossby wave mode. Note here also that the resolution of the LSP (FWHM) which is about 1/T (see Cumming et al. , 1999), where T is the length of the time window (in this period range 25 d). This means that single observations with periods of 15 d or 16 d are not significantly different.

Another wave that has frequently been mentioned in literature is the quasi-5-day wave (e.g. Wu et al., 1994; Jarvis , 2006; Day

and Mitchell, 2010a). These observations are very likely observations of the Rossby wave (1,1) mode (Kasahara , 1980; Salby , 1981b). In our observations we observe also a larger number of significant events in this period range. More precisely, we observe two peaks, one at about 5 d and one at about 6 d. However, due to the resolution of the LSP, these two peaks would





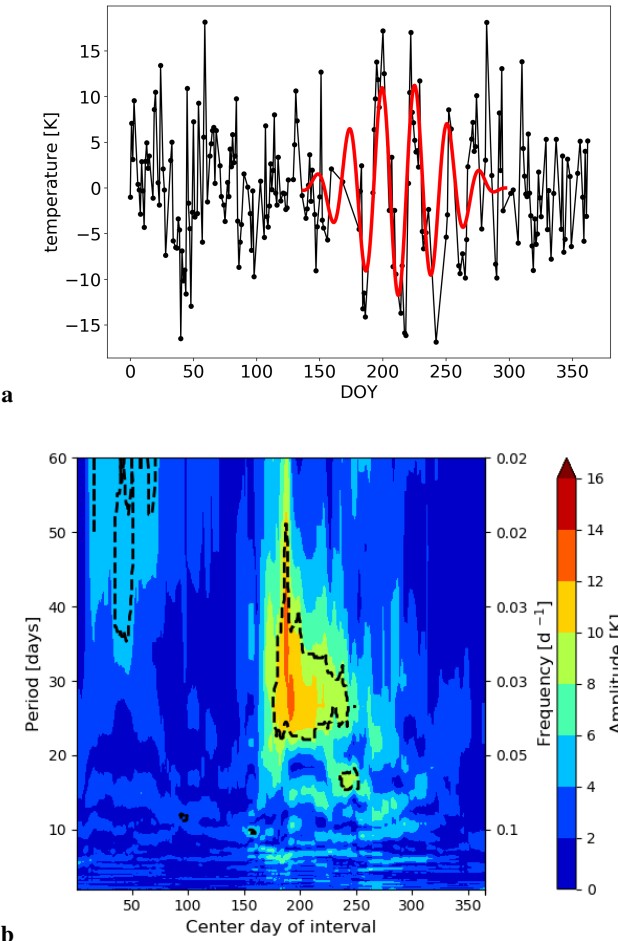

**Figure 7.** Example for an event with a large temperature fluctuation with a period of about 28 d. **a** The temperature time series after subtracting the seasonal variations is shown as black curve. The red curve is the fit to the data in the middle part of the time series. The x-axis show days of the year (DOY) with start date July 1, 1997. **b** LSP result for the amplitudes of periodic fluctuations over the winter 1997/1998. The x-axis show the centre days of the windows with start date January 1, 1997 and the y-axis show the period and frequency of the fluctuations, respectively. The amplitude is displayed colour coded.

not be significantly different for single detections. The predicted range of periods for the observation of the Rossby wave (1,1) mode is reported to be 4.4 to 5.7 d (Salby , 1981b). This range is fitting quite well to our observations which makes us believe that these period range shows observations of this specific Rossby wave mode. The quasi-10-day wave is also a known wave type in the atmosphere (e.g. Jarvis , 2006; Takahashi et al., 2013; Forbes and Zhang , 2015). It is likely the manifestation of the Rossby wave (1,2) mode (Kasahara , 1980; Salby , 1981b; Forbes and Zhang , 2015). The period range of predictions goes from 8.3 to 10.6 d (Salby , 1981b). In our observation we observe a very prominent peak at about 10 d and two additional



peaks nearby at slightly above 8 d and slightly below 12 d. According to the resolution of the LSP results and with respect to
the potential period range of those waves we believe that, at least the largest part, of these observations show the Rossby wave
(1,2) mode. The last major part of events is detected in the period range above 30 d. As mentioned above, the large increase
is at least partly caused by the fact that periodic fluctuations with longer periods typically also lasts longer which enhances
the days at which these waves are observed. Nonetheless, it does not explain the observation itself. Planetary waves with very
similar periods have also been reported for other observations sites in Antarctica (Espy et al., 2005; Stockwell et al., 2007).
Thus, our observations are likely also planetary waves of wavenumber 1. A better constraint of the origin of the waves and
their complete strucucture would need additional data and further investigations which is beyond the scope of this paper.

The seasonal variation of the wave activity is different for different type of waves. In the period range below 10 d we observe
two maxima in the activity around the spring equinox in April and in late summer (August/September) (compare Fig. 4). In the
case of waves with longer periods ($> 20$ d) a large maximum in winter and late fall is observed whereas the activity in summer
and other seasons is comparable small (compare Fig. 4). A larger wave activity for waves with smaller periods in other seasons
than winter has also been observed by others. Wu et al. (1994) analysed two years of HDRI (High Resolution Doppler Imager)
data and observed wave events of the quasi-5-day wave mainly in April/May and in September to October. Radar observations
over India showed larger activities of the 6.5-day wave around the equinoxes (April/May and September/October) (Kishore et
al., 2004). Riggin et al. (2006) also observed larger events in April/May when analysing SABER data over a time period of
three years. Observations in both hemispheres at polar sites showed strong wave activity for the 5-day wave in winter and late
summer (August/September) but no significant events at equinoxes (Day and Mitchell, 2010a). In a former study of the GRIPS
observations at Wuppertal the authors also reported a larger amount of events in summer time in the case of planetary waves
with smaller periods (Bittner et al., 2000). In total, the seasonal variation of the wave activity in the case of smaller periods
observed at Wuppertal agrees quite well with other observations. The observation of large planetary wave events in summer
cannot be explained with the excitation of these waves in the troposphere and their propagation up to the MLT region, because
the mean stratospheric flow in summer prevents these upward propagation of most waves (e.g. Charney and Drazin , 1961).
Thus, other mechanism are necessary to explain the observations. Two main mechanism are discussed in previous work. On
the one hand, the waves can be exited at higher altitudes and propagate upwards afterwards and, on the other hand, ducting
from the winter hemisphere to the summer hemisphere could take place (e.g. Bittner et al., 2000; Riggin et al., 2006; Day and
Mitchell, 2010a, and references therein). In the period range between 10 and 20 d we observe activity in most of the seasons
except for summer. This can be seen in difference between the lower middle panel and the lower left panel of Fig. 4. A clear
enhancement of the activity is observed during the whole year except for the summer months. The quasi-16-day wave and
a part of the observations of the quasi-10-day wave are mainly responsible for this enhancements. Most of the other studies
found in literature also report on similar seasonal distributions. For the quasi-10-day wave Forbes and Zhang (2015) presented
maximum activity in winter and around the equinoxes in mid-latitudes. In the case of the quasi-16-day wave the picture is
quite similar with larger activity in winter and only minor activity in summer months (Luo et al., 2000; Day and Mitchell,
2010b). Nonetheless, the quasi-16-day wave has also been observed in summer (e.g. Espy et al., 1997). Discussed mechanism
are again local phenomena and ducting from the winter to the summer hemisphere (see Espy et al., 1997; Luo et al., 2000, and





references therin). In the case of wave events with periods larger than 20 d we observe a large summer to winter differences
with a huge number of events that took place in winter time or late fall and only a very small number of events in summer.
As already mentioned this is expected because of the wave filtering in summer that prevents the wave propagation to the MLT
region. These findings are also confirmed by previous other studies. The observation of the large 28 d wave event reported by
Zhao et al. (2019) also took place in winter. Similarly, Stockwell et al. (2007) report that the largest activity of waves in the
period range from 30 to 50 d is observed in late winter.

**4.2   Long-term behaviour of wave activity**

We see a main long-periodic fluctuation with a period of about 20 years in the long-term behaviour of the wave activity. This
can be seen in two of the three different proxies. Of the two new proxies only the yearly mean amplitude of the significant
events shows a clear long-term behaviour. Therefore, the standard deviation, which is mainly determined by the amplitude of
the significant fluctuations in most years, also shows a very similar long-term behaviour. The fact, that the time series of the
weighted sum of days with significant results does not show the same long-period fluctuation suggests that in years with higher
activity during the quasi-bidecadal oscillation (for example mid-1990s) not more events are expected but events with larger
amplitudes than in the years with lower activity (for example around 2005).

The quasi-bidecadal oscillation of the standard deviation has also been observed by Höppner and Bittner (2007) for a shorter
time interval of observations (until 2005). Thus, their findings agree well with ours and the wave activity observed by the
standard deviation proxy proceeds still in this quasi-bidecadal oscillatory way. Jarvis (2006) also observed a quasi-bidecadal
oscillation in the wave activity of the 5-day planetary wave. Like us Jarvis (2006) observed the quasi-bidecadal oscillation
in the residual planetary wave amplitudes. The observed change lies in the range of about 10% which is in line with our
observations here, where we see a change of roughly ±1 K and a mean amplitude of about 8 K. In contrast to our findings,
where we see the quasi-bidecadal oscillation in the mean over all periods in the range from 2 to 60 d, Jarvis (2006) observed
the long-term fluctuation for the 5-day planetary wave only and not for the 10-day and 16-day planetary wave.

In former studies of the OH*(3,1) rotational temperatures we already observed the quasi-bidecadal oscillation in yearly mean
temperature observations (Kalicinsky et al., 2016, 2018, 2024). The phase of the quasi-bidecadal oscillation of the temperature
observations is nearly the same as that of the mean amplitude and the standard deviation of the residuals, respectively. All of
the time series show maxima around the mid-1990s and around 2015/16 in addition to a minimum around 2005. This means the
larger amplitudes of the significant wave events occur together with enhanced yearly mean temperatures and vice versa. As we
also observed the quasi-bidecadal oscillation in other altitudes such as the mesosphere and stratosphere with alternating sign
from one atmospheric layer to the other (stratosphere and lower thermosphere (GRIPS OH*(3,1) rotational temperatures) are in
phase and mesosphere is shifted by 180°) (Kalicinsky et al., 2018) and also on ground (Kalicinsky and Koppmann , 2022), this
temperature oscillation in the atmosphere might influence other atmospheric parameters such as the wind and, therefore, also
influence the wave filtering. Likely, only in certain years the conditions are favourable for waves with smaller amplitudes to
reach higher altitudes, whereas in other years only waves with amplitudes that are large enough reach the observation altitude



of the GRIPS instrument. But a complete analysis of this hypothesis will require additional data is beyond the scope of this paper.

## 5  Summary and conclusions

We analysed more than 30 years of OH(3,1) rotational temperatures observed from Wuppertal with respect to periodic fluctuations in the period range from 2 to 60 d. Fluctuations with a period around 28 d are the main fluctuation observed in the last decades. These fluctuations are most likely Rossby waves (1,4) mode that appear with a period of about 28 d in ground-based measurements in presence of a zonal background wind. Other period ranges which are often detected in the observations are 5 to 6 d, 8 to 12 d, and around 15 d. These period ranges likely can be assigned to the Rossby wave (1,1), (1,2), and (1,3) mode,
respectively.

Most of the wave activity is observed in winter time because of the different wave filtering in summer and winter, i.e. the conditions in winter are typically more favourable for waves to reach higher altitudes. This winter to summer difference is not universal for all waves. It holds for waves with larger periods, but it breaks off in the case of smaller periods below about 20 d. These waves with smaller periods occur more evenly distributed across the year with even a ´larger number of events around
the equinoxes.

The mean amplitude of all observed significant events is about 8 K, whereby the amplitudes range from about 2 to 16 K. A significant difference for the distribution of the amplitudes dependent on the period range cannot be observed.

The analysis of the long-term behaviour of the wave activity revealed a quasi-bidecadal oscillation. This oscillation is observed in two wave proxies, the standard deviation of the temperature residuals and the mean amplitude of the significant events within
a year. Since the last wave proxy, the amplitude weighted sum of days with significant results, does not show this fluctuation, the conclusion is that the amplitude is the quantity that shows the quasi-bidecadal oscillation. This means, that in certain years not more events but events with larger amplitudes are expected, whereas in other years the mean amplitude of the events is smaller. The quasi-bidecadal oscillation is in phase with the oscillation of the yearly mean temperatures themselves. A connection between changes in the background temperature field and, thus, also other parameters, may influence the wave filtering
and lead to the observation of the quasi-bidecadal fluctuation of the wave activity

*Data availability.*  The OH(3,1) rotational temperatures which were derived from the GRIPS observations at Wuppertal can be obtained by request to the corresponding author.

*Author contributions.*  CK conceptualised the study. RR performed the analyses of OH(3,1) rotational temperatures under intensive discussion with CK. PK provided the OH(3,1) rotational temperatures. The article was written by CK with contributions from all coauthors.





*Competing interests.* The authors declare that they have no conflict of interest.

*Acknowledgements.* This research was funded by the Deutsche Forschungsgemeinschaft (DFG, German Research Foundation) – 519284835.



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
