# Peer review of "Ground-based observations of periodic temperature fluctuations in the mesopause region with periods larger than 2 days"

_EGUsphere, 2025_

## Author Comment (AC1)

**Reply to the comments on the manuscript**

"Ground-based observations of periodic temperature fluctuations in the mesopause region with periods larger than 2 days"

by Christoph Kalicinsky, Robert Reisch, and Peter Knieling

We thank the reviewers for their helpful comments and recommendations. In the following, we discuss the issues addressed by the reviewers and explain our opinions and the modifications of our manuscript.

We enumerate the comments and repeat them in bold face. The modifications of the manuscript are displayed in the marked-up manuscript version as coloured text. Deleted parts are shown in red and new or modified text parts in blue.

**1 Comments Reviewer 1**

The paper "Ground-based observations of periodic temperature fluctuations in the mesopause region with periods larger than 2 days" by C Kalicinsky et al., utilized more than 30 years of mesospheric temperature observations to study the oscillations and their long-term variations. Lomb-Scargle periodogram (LSP) is used to identify the periods of these oscillations. Fluctuations with periods of 5/6 days, 8-12 days, 15 days and 28 days were identified and related to Rossby waves. Most of the activities occurred during the winter season for longer period waves, while the short period waves peaked during equinoxes. The authors claim that long-term variations of the wave activity showed a quasi-bidecadal signature. This is a very valuable study and could provide crucial information about planetary waves and their long-term behavior. However, I have some concerns that need to be addressed.

Major concerns:

1. I can't agree with the way the authors counting wave events. By doing this, it makes the results in Figure 3 misleading and hard to understand. I could not tell how many times the waves in each bin had happened during this >30 years period. For example, the 28 day wave event claimed to be the most popular one, it has the same count as the 45-60 day waves, 350. What does it mean? Which one occurred more often by how many times? By looking at the duration of the 28 day wave in Figure 7 ( $\sim$ 70 days), does it mean only  $\sim$ 5 times this wave occurred during the >30 years of observation? From the spectrum results of Figure 2 and Figure 7, one can see that within one year, there were not a lot of wave events identified which is totally normal.

It is more reasonable to count each event as one which will clearly show how many times each wave happened throughout this very long data set.

We agree that the way we count the wave events can be improved. We added a second version of the histogram. The first one is kept as before. The second one is now a histogram where the counted significant days are normalized by the period in the corresponding bin. This was suggested by reviewer 2. Thus, the new statistic measures the importance of a wave type in terms of cycles and it is now independent of the period

of the wave itself. As a consequence the increase of the values in the histogram with increasing period is removed and the peaks at smaller periods are enhanced showing the relative importance of these wave types. We believe that this way to count the events is slightly better as it still takes into account the difference between short and long events and does treat them in the same way even if one event includes one cycle and the other several cycles.

However, we still keep the former version with the significant days as it also contains information on the importance of the different waves with respect to the total time period in which they were observed. In total, it is clearly visible that the 28 day wave is the most important one, as it is observed at the largest number of days and it is also the wave event where the most cycles have been observed. Additionally, at a period of about 2 days another type of wave has a somehow larger peak in the histogram with the normalized bins. We added all new information to the corresponding paragraphs.

**General comments:**

- 1. The period range in Figure 4 is confusing with each subset inclusive of the others. In the top row, the difference between period >10 days and period >20 days are small, which indicates the waves with periods between 10 and 20 days, does not occur a lot. Does this mean the quasi-16 day waves only show up in your data very occasionally? Again, the counting mechanism make it hard to understand the seasonal variations of the detected wave events. Independent spectrum range would help with the results.
  - We updated the plot and removed 4 of 6 subfigures. Finally, we only kept the results showing the seasonal variation of the significant days for periods smaller than 20 days and larger or equal than 20 days. Thus, we divided the results in two parts only. The waves with larger periods clearly show the largest number of counts in winter whereas the waves with smaller periods show maxima at equinoxes. Thereby, the waves with periods below 10 days account for the major part of observations in summer and around equinoxes. We rephrased the corresponding parts of the text to explain such details. Indeed, the quasi-16 day wave plays only a minor role as can be seen in the new plot where the histogram values are normalized by the periods of the corresponding bins (See
  - where the histogram values are normalized by the periods of the corresponding bins (See major concerns bullet 1). In the new figure 4 we kept the counting mechanism as we think that the number of days at which significant events occur during the year clearly shows the differences between summer and winter for the different periods. When only the number of events would be counted a e.g. 10-day event in summer would have the same weight as a 45-day event in winter.
- 2. Section 3.2 focused more on the similarities and differences among the 3 quantities. It is not clear how introducing the 2 new proxies, especially the later one would help in drawing a clear conclusion. In Figure 6b, only the ~20 year variation was mentioned, what about the shorter periods with even stronger power? How significant of this 20 year period oscillation? What is the confidence level for this result?
  - In the former study by Höppner and Bittner (2007) the standard deviation was used as proxy for the wave activity. This has the drawback that all kind of fluctuations are included in this quantity and not only the significant ones. Because of this, we introduced the mean amplitude of the significant events as a new proxy. This also

has a small drawback, because it does not include the length of the events, only the strength. Thus, we also introduced a third proxy which is the amplitude weighted sum of significant days. As all of the three proxies have similarities and also differences the comparison of the individual LSPs can help to gain information on the importance of the length and the strength of the wave events on certain periodic behaviours. As the standard deviation mainly depends on the amplitude the LSPs of the standard deviation and the mean amplitude look quite similar. This means that the long-periodic fluctuation at about 20 years is likely caused by events with larger amplitudes in some years and smaller amplitudes in other years instead of longer or shorter events. In the case of the 4-year oscillation this is different. This oscillation is seen in all proxies but the largest power is seen for the weighted sum of days. This would suggest that the length of significant events shows a quasi-quadrennial oscillation.

The two main peaks of the LSP for the standard deviation at about 20 years and 4 years are not significant at a 95% confidence level with respect to the complete analysed period range (FAP). This is a common problem with this rather conservative approach in the case when several similar large peaks occur in a periodogram, which means that a larger number of oscillations shares the almost total variance of the complete time series, and does not mean that the oscillations are not real. With respect to the single frequency the significant level for the peak at 20 years is almost 95%, i.e. it is rather uncertain that a peak with that height occurs at exactly this period just by chance. As we are searching for a peak at a period of 20 years because of the former study by Höppner and Bittner (2007), this second way is valid in our study here.

3. The LSP results for 28 day waves in Figure 7b does not make sense comparing with the data in Figure 7a. During the event of the 28 day wave, the spectrum results showed a very broad peak (period extended from 25 to nearly 50 days) while the data (Figure 7a) showed a highly defined wave period which would be a narrow horizontal maximum in Figure 7b. With such a broad peak, how the period of 28 days is concluded? The other thing of this 28 day wave event is it happened in July and August, which is summer for the northern hemisphere which is contradict to what the authors conclusion (line 240).

First, there is a mistake in the caption of the figure. Both subfigures a and b show the time period from July 1st to June 30th and, thus, they show the same event that took place in winter. We rephrased the caption.

At the beginning the peak is very broad and the period of the maximum lies slightly above 30 days, but after this short time period the maximum lies in the region between 25 to 30 days. The mean period of the fit shown with the red curve in Fig 7a is about 26 days. Therefore, the majority of the observations will be counted in the maximum bin from about 25 to 30 days (compare Fig. 3) and we denoted it as a quasi-28 day wave event.

We rephrased the corresponding parts of the text and added more details.

4. In general, observations from one ground-based site are not enough for planetary wave mode identification. Section 4.1 tried to relate the observed periods of oscillations to known Rossby waves of certain mode. The Rossby mode has certain latitudinal and longitudinal structures. Relating waves observed from different latitudes can lead to wrong conclusions. Also, for the waves of 4-6 days, 16 days, most of the studies cited were using observations of the mesospheric wind. The normal modes of winds are quite different from the ones for temperatures.

We agree that you need additional data to identify the wave modes. We rephrased our statements according to this in Section 4.1.

**Minor comments:**

We changed the manuscript according to the minor comments where still necessary.

---

## Author Comment (AC2)

**Reply to the comments on the manuscript**

"Ground-based observations of periodic temperature fluctuations in the mesopause region with periods larger than 2 days"

by Christoph Kalicinsky, Robert Reisch, and Peter Knieling

We thank the reviewers for their helpful comments and recommendations. In the following, we discuss the issues addressed by the reviewers and explain our opinions and the modifications of our manuscript.

We enumerate the comments and repeat them in bold face. The modifications of the manuscript are displayed in the marked-up manuscript version as coloured text. Deleted parts are shown in red and new or modified text parts in blue.

**1 Comments Reviewer 2**

**General comments:**

This manuscript presents an analysis of seasonal and long-term periodicities in 30 years of mesopause region temperatures derived from hydroxyl (3-1) band spectrometer observations from Wuppertal, Germany. They use a variable sized, sliding window to apply a Lomb Scargle periodogram analysis to deseasonalized time series for each year. They analyse the main seasonal periodicities in the range of 2 to 60 days in terms of planetary Rossby wave activity. The majority of wave activity is observed in winter, with some indication of a maximum occurrence of shorter period waves at the equinoxes. Long-Term behaviour of wave activity was analysed using the standard deviation of temperature residuals and the mean amplitude of significant events as proxies. This revealed a quasibidecadal oscillation characterised by events with larger amplitude. The work is a worthwhile study on planetary wave activity in one of the longest hydroxyl rotational temperature data sets and yields some sensible but not unexpected results in terms of predominant wave activity. Some further work and revisions are suggested below.

Specific comments:

1. Line 14 Introduction section: In the context of Lomb Scargle analyses of long term hydroxyl rotational data sets the work from the Davis, Antarctica observatory, seems particularly relevant (for example French and Klekociuk (2011) https://doi.org/10.1029/2011jd015731, French et al (2020) https://doi.org/10.5194/acp-20-8691-2020) but is not considered in the introduction or in the discussion on periodic fluctuations in 4.1. Including a comparison with these studies would be valuable, particularly since similar periodicities from other Antarctic sites are already discussed in the periodic fluctuations section (e.g. line 279)

We indeed missed to mention this important work on the time series of OH\* temperatures from Davis, Antarctica. We included the publication that partly deals with planetary wave activity by French and Klekociuk (2011) in the introduction and discussion part and the corresponding positions. We further included the study by French et

al. (2005) where a detailed examination of a 14-day wave is done.

The study by French et al. (2020) is considered in the discussion of the long-term evolution. See point 10.

2. Line 85 is there much variation in the amplitude and phase of the seasonal fit between each year? What are the ranges of the fit coefficients? What is the difference to a seasonal fit to all years?

There are some variations of the fit parameters, e.g. there is a declining trend of the amplitude of the annual cycle and some further fluctuations from year to year. A fit to all years at once would not capture all of these year to year fluctuations and the long-term variations. Thus, the residuals after subtraction of such a fit would still contain some variations with implications on further analyses. We added some more information.

3. Line 100-105 why not do a sliding window LS over the entire deseasonalized time series instead of adding 2 months to either side of each year? This paragraph also needs revising eg "all possible time windows have been of the complete time series were analysed" makes no sense.

See point 2. One fit for all years would not be able to capture all changes in the seasonal behaviour in each year. As a consequence the fit would be not precise enough in each single year and there would be some shifts of the residuals (when the fit lies slightly above or below the best possible fit) and even long-term trends that remain in the residuals in some years (when the fit changes from below to above or vice versa). This would result in wrong or biased LSP results. We added the 2 months at each sides to be able to calculate a LSP for each possible day of the whole time series of more than 35 years and to account for all of this year to year variations of the seasonal cycle. After a deseasonalization of each single year one could stack a individual residuals together and calculate the LSPs, but this would not lead to different results. We added more information in the paragraph.

- 4. The paragraph explaining the variable window length (lines 117-123) is incoherent and difficult to read and interpret. This should be revised for clarity. We rephrased the paragraph.
- 5. The "factor" introduced in line 126 .. should be defined as cycles per window length of the period being analysed. How is this used in the analysis except to step up the window length for increasingly long periods analysed? We already described that the factor can be seen as cycles per window length. We rephrased the paragraph for clarity.
- 6. Line 163 What is the criteria for significance here and in Fig 2 and how is this computation affected by the varying window lengths?

The significance of the results is analysed using the false alarm probability (FAP). The FAP gives the probability that a certain peak can occur just by chance somewhere in the analysed period range. It is determined using an empirical expression where the coefficients were determined by using Monte Carlo simulations. By using this expression the significance for each peak can be evaluated depending on the window length, period range and number of data points (for more information see Kalicinsky et al. 2020). We added this additional information to section 2.3 and section 3.

7. Section 3.1 and Fig 3. Occurrence frequencies are counted by the centre days which meet the significance criteria. This will bias occurrence statistics to waves of longer duration. The occurrence of long period waves will necessarily occur on a larger number of days (and this is stated in Line 172). I don't think this is a valid assessment of wave occurrence statistics and the analysis needs to be revised. Normalise by the period length or count waves of each period by event.

We added a second version where we normalize the histogram. (See also Reviewer 1)

- 8. Figure 4. This is a confusing plot as the period ranges overlap. Why not separate these ranges and do periods <10, 10-20, 20-30 and >30?

  We reduced the plot to 2 subfigures only and added some additional information in the text (see Reviewer 1).
- 9. Line 192 and Fig 5. In the discussion of observed amplitudes what is the effect of the integration of the temperature variation by a wave propagating through the 8km layer profile of the hydroxyl emission?. Does this limit the observable frequency and amplitude of waves.
  - There might be some reduction of the amplitudes due to vertical averaging, especially for waves with smaller periods and smaller vertical wavelengths. In the majority of the cases waves with periods even below about 10 days exhibit vertical wavelengths that are a multiple of the FWHM of the OH layer (e.g. Buriti et al., 2005, Ern et al., 2013, Yamazaki and Matthias, 2019, Reisin, 2021). Very small vertical wavelengths below e.g. 20 km are only reported for a very minor portion of all observations (Reisin, 2021). Thus, the averaging effect should be small or maybe even negligible.
- 10. Section 3.2 and Figure 6. In the discussion of long-term wave activity the clearly dominant periodicity in the weighted sum power spectrum at 4 years is not discussed. This has higher power than any other measure including the bi-decadal oscillation. What is this attributed to and does it align with the quasi-quadrenial oscillation of French et al (2020) https://doi.org/10.5194/acp-20-8691-2020?

The significance of the 4-year peak is only good with respect to the single frequency not with respect to the whole frequency range. However, as the peak occurs in all LSPs and is in the case of the weighted sum the highest in all of the periodograms, we analysed it further. It looks quite similar to the one reported by French et al. (2020) concerning the period (4 years here compared to 4.2 years) and the phase, where there seems to be a slight time lag of about 1 year with the Wuppertal oscillation preceding. We included this information to the manuscript.

**Minor corrections:** We changed the manuscript according to the minor comments where still necessary.